# Effects of Eicosapentaenoic Acid on Arterial Calcification

**DOI:** 10.3390/ijms21155455

**Published:** 2020-07-30

**Authors:** Yukihiro Saito, Kazufumi Nakamura, Hiroshi Ito

**Affiliations:** Department of Cardiovascular Medicine, Okayama University Graduate School of Medicine, Dentistry and Pharmaceutical Sciences, Okayama 700-8558, Japan; gsaito8976@gmail.com (Y.S.); itomd@md.okayama-u.ac.jp (H.I.)

**Keywords:** eicosapentaenoic acid, atherosclerosis, Klotho

## Abstract

Arterial calcification is a hallmark of advanced atherosclerosis and predicts cardiovascular events. However, there is no clinically accepted therapy that prevents progression of arterial calcification. HMG-CoA reductase inhibitors, statins, lower low-density lipoprotein-cholesterol and reduce cardiovascular events, but coronary artery calcification is actually promoted by statins. The addition of eicosapentaenoic acid (EPA) to statins further reduced cardiovascular events in clinical trials, JELIS and REDUCE-IT. Additionally, we found that EPA significantly suppressed arterial calcification in vitro and in vivo via suppression of inflammatory responses, oxidative stress and Wnt signaling. However, so far there is a lack of evidence showing the effect of EPA on arterial calcification in a clinical situation. We reviewed the molecular mechanisms of the inhibitory effect of EPA on arterial calcification and the results of some clinical trials.

## 1. Introduction

Arterial calcification is often observed in patients with atherosclerosis, chronic kidney disease (CKD) and diabetes mellitus. Coronary artery calcification detected by computed tomography (CT) reflects atherosclerotic plaque burden and predicts cardiovascular events [1,2]. Additionally, aortic calcification predicts mortality and nonfatal cardiovascular events in dialysis patients [3]. However, macrocalcifications occurring in the intimal plaque during late-stage calcification are believed to stabilize the plaque [4]. On the other hand, microcalcifications occurring during the early-stage calcification correlate with plaque vulnerability that results in myocardial and cerebral infarction, but it is difficult to detect microcalcifications using a clinical CT scan [5]. Medial calcification causes arterial stiffness that is a predictor of coronary heart disease and stroke [6,7].

Lowering low-density lipoprotein-cholesterol (LDL-C), which is an established therapy for cardiovascular diseases, promotes formation of coronary artery macrocalcifications [8,9,10]. Although there is no clinically established therapy for arterial calcification, some molecules are expected to reduce arterial calcification, for example, eicosapentaenoic acid (EPA), vitamin K1, magnesium, spironolactone and evolocumab [11,12,13,14,15,16].

Purified EPA has been prescribed to lower triglycerides in Japan. EPA inhibits osteoblastic change and mineralization in vascular cells [17]. Similarly to Kanai et al., we found that EPA significantly suppressed arterial calcification in experimental studies using small animals [18,19]. Thus, we expect that lowering arterial calcification by EPA leads to a reduction of cardiovascular events. Although docosahexaenoic acid (DHA) also inhibits calcification of vascular cells in vitro, it raises serum LDL-C levels, which could be a risk of cardiovascular diseases [17,20,21].

In the Reduction of Cardiovascular Events with Icosapent Ethyl-Intervention Trial (REDUCE-IT) and the Japan EPA Lipid Intervention Study (JELIS) using highly purified EPA, EPA significantly reduced the risk of cardiovascular events [22,23,24]. However, the effect of EPA on arterial calcification in a clinical situation is not established. While the blood level of omega-3 fatty acid was shown to be inversely associated with coronary artery calcification in some observational studies, pitavastatin plus EPA did not reduce the progression of coronary artery calcification compared with the effect of pitavastatin alone in patients with hypercholesterolemia [25,26,27,28]. Since there is a discrepancy in the effect of EPA among the results of clinical observational, clinical interventional and experimental studies, we reviewed the effect of EPA on arterial calcification.

## 2. EPA

EPA is a polyunsaturated long-chain fatty acid found in fish oil with a 20-carbon backbone and five cis-double bonds at positions 5, 8, 11, 14 and 17. During an inflammatory response, arachidonic acid and EPA are metabolized by cyclooxygenases and lipoxygenases to form eicosanoids. Eicosanoids derived from EPA, prostaglandin-3 family, thromboxane-3 family and leukotriene-5 family are less potent inducers of inflammation, blood vessel constriction, and thrombus formation than eicosanoids derived from arachidonic acid, prostaglandin-2 family, thromboxane-2 family and leukotriene-4 family. Additionally, E-series resolvins are attractive metabolites derived from EPA that work as specialized pro-resolving mediators [29,30].

## 3. Effect of Highly Purified EPA on Cardiovascular Disease

Statin therapy for lowering LDL-C decreases the risk of coronary heart disease and mortality [31]. However, residual risk persists even after the achievement of target LDL-C levels. Low levels of high-density lipoprotein-cholesterol and high level of triglycerides are residual risks that cannot be controlled by LDL-C lowering therapy with statins. However, previous trials with fibrates or niacin for lowering triglycerides failed to reduce events after LDL-C levels were controlled with statins [32,33,34]. Additionally, n-3 fatty acid products were not beneficial for patients receiving statin therapy in some previous trials [35,36,37]. On the other hand, treatment with 1.8 g/day purified EPA plus statin therapy resulted in a 19% relative risk reduction in major coronary events in the JELIS (statin plus EPA, 2.8% vs. statin alone, 3.5%) [23]. However, that study was a non-blinded randomized controlled trial (RCT). 

The REDUCE-IT, a blinded RCT, also demonstrated that 4 g/day purified EPA reduced cardiovascular events: a composite of cardiovascular death, nonfatal myocardial infarction, nonfatal stroke, coronary revascularization and unstable angina for EPA vs. placebo was 17.2% vs. 22.0% [22,24]. However, the cardiovascular risk reduction was not associated with attainment of more normal triglyceride levels in REDUCE-IT [22]. These results suggest that lowering triglycerides does not mainly contribute to the risk reduction and that there is another specific target of highly purified EPA. Actually, various beneficial effects of EPA on atherosclerosis have been reported [38]. The serum level of small dense LDL, especially common in the serum of patients with atherosclerosis and susceptible to chemical modifications increasing its atherogenicity, was reduced by 1.8 g/day EPA, indicating that EPA improves lipoprotein profiles [39,40]. Additionally, EPA was preferentially incorporated into the thin-cap plaque rather than the thick-cap plaque and reduced atherosclerotic lesions in some mouse models. Atheroma formation in ApoE-deficient mice and LDL receptor-deficient mice and aortic aneurysm formation in osteoprotegerin-deficient/ApoE-deficient mice were prevented by orally administered EPA through anti-inflammatory effects [41,42,43,44].

## 4. Arterial Calcification in Cardiovascular Disease

### 4.1. Arterial Calcification and Clinical Prognosis

Arterial calcification is a hallmark of advanced atherosclerosis. Standard coronary risk factors are related to both the incidence and progression of coronary artery calcification [45]. An annual calcium score progression of >15% is associated with worse mortality [46].

Pathomorphologically, arterial calcification can be divided into two distinct entities according to the calcified sites within the arterial wall: patchy calcification of the intima close to lipid deposits as present in plaque calcification and calcification of the media without lipid deposits, known as Mönckeberg-type [47]. Both types are often observed in patients with cardiovascular disease induced by diabetes mellitus, metabolic syndrome, CKD, and aging [48].

Arterial calcification evaluated by CT is a predictor of cardiovascular events [2,49,50,51]. There are several biological mechanisms by which arterial calcification increases cardiovascular mortality. Coronary arterial intimal calcification, particularly microcalcification within the fibrous cap, is thought to increase the risk of plaque rupture and thrombus formation resulting in myocardial infarction [52,53]. However, it is difficult to see microcalcification by a clinical CT scan due to its low spatial resolution. On the other hand, macrocalcification can be identified using CT and is widely believed to stabilize the plaque [54,55]. Arterial medial calcification increases arterial stiffness, which causes systolic hypertension, diastolic dysfunction, decreased coronary perfusion and heart failure [50,56,57]. 

### 4.2. Arterial Calcification and Drug Therapy for Cardiovascular Disease

Several recent studies with large sample sizes have suggested that statins promote vascular calcification, even though LDL-C-lowering treatment with statins is an established drug therapy for patients with coronary artery disease [8,9,58]. That is, statins regress atheroma volume but promote macrocalcifications. From these findings, progression of macrocalcifications by statins may be one of the mechanisms by which vulnerable plaque is stabilized [59]. Atorvastatin increased atherosclerotic plaque calcification also in ApoE^−/−^ mice [60]. Statins activate Rac1 in macrophages, resulting in Rac1-dependent interleukin 1β upregulation and consequent calcification [60,61,62]. 

Fibrates and niacin, other lipid-lowering drugs, do not suppress the progression of arterial calcification caused by statins. Interestingly, evolocumab, a proprotein convertase subtilisin-kexin type 9 (PCSK9) inhibitor, could suppress the increase in coronary artery calcification caused by statins [14]. Plasma PCSK9 concentration has been shown to be associated with coronary artery calcification in untreated patients with angina-like chest pain [63]. Statins decrease intracellular cholesterol in the liver and increase PCSK9 as well as LDL receptors [64]. Therefore, a PCSK9 inhibitor might cancel the action of PCSK9 increased by statin treatment and prevent arterial calcification. However, another study did not detect the inhibitory effect of evolocumab on calcification [65].

## 5. Molecular Mechanisms of Arterial Calcification

Although arterial calcification initially seemed to be a passive process resulting from oversaturation of plasma with calcium and phosphate, it has become clear that it is a highly regulated process involving several cell types and molecular mechanisms [66,67]. While inflammation and oxidative stress generally cause intimal calcification [68], uremia and high serum levels of calcium and phosphate drive medial calcification [69]. Additionally, complex mechanisms are common in both intimal calcification and medial calcification. The mechanism included failed anti-calcification processes due to loss of calcification inhibitors including osteoprotegerin, osteopontin, matrix Gla protein (MGP), fetuin-A and pyrophosphate [48,70,71,72,73], induction of osteo/chondroblast-like cells producing extracellular vesicles [74], cell death resulting in release of apoptotic bodies or necrotic debris that cause nucleation of apatite [75], calcium/phosphate dysregulation causing deposits calcium phosphate hydroxyapatite [76], nucleation complexes formed during bone remodeling, and matrix degradation/modification [77].

## 6. Inhibitory Effects of EPA on Arterial Calcification in Experimental Studies

Several experimental studies have demonstrated that EPA suppresses arterial calcification in vitro and in vivo via various mechanisms (Table 1). MCP-1, monocyte chemotactic protein-1; MMP, matrix metalloproteinase; NOX, NADPH oxidase; GPR120, G-protein coupled receptor 120; PPARγ, peroxisome proliferator-activating receptor gamma; NF-κB, nuclear factor-κB; ACSL3, Acyl-CoA synthetase long chain family member 3.

### 6.1. Warfarin-Induced Arterial Calcification

Warfarin antagonizes vitamin K and blocks gamma-carboxylation of MGP and induces vascular calcification. MGP has been identified as a vitamin K-dependent protein and a calcification inhibitor in cartilage and the vasculature. MGP-deficient mice develop widespread aortic and arterial calcification [80,81]. MGP is an independent predictor of both intimal and medial vascular calcification in CKD [82]. It binds to bone morphogenetic protein (BMP)-2, triggering the transformation of vascular smooth muscle cells to osteoblast-like cells, and inhibits the osteoinductive effect of BMP-2 [83,84]. The function of MGP depends on its gamma-carboxyglutamic acid residues. To modify the residues, vitamin K is required as a co-factor [85,86,87,88]. 

Kanai et al. demonstrated that EPA prevents warfarin-induced arterial medial calcification in rats [19]. EPA suppresses osteogenetic marker expression, macrophage infiltration, matrix metalloproteinase (MMP) activity and monocyte chemotactic protein-1 (MCP-1) expression in the aorta.

### 6.2. Hypomorphic Klotho-Induced Arterial Calcification

Klotho (α-Klotho) was identified as an anti-aging hormone in mice [89]. Klotho is a coreceptor of fibroblast growth factor 23 and regulates calcium/phosphate metabolism and vitamin D synthesis. Klotho overexpression extends the life span in mice [90]. Hypomorphic mutant Klotho [kl/kl] mice exhibit accelerated aging phenotypes including arteriosclerosis, ectopic calcification, osteoporosis, pulmonary emphysema, muscle atrophy and short life span [91]. Extensive medial calcification is observed in the aorta and middle-sized muscular arteries. The vascular changes seen in kl/kl mice are similar to Mönckeberg-type arteriosclerosis in humans. Also in humans, low serum KLOTHO levels are associated with the prevalence of cardiovascular disease and all-cause mortality [91,92]. Furthermore, KL mRNA and KLOTHO protein levels decrease in the kidneys of patients with chronic kidney disease and there is an inverse association between serum KLOTHO level and arterial stiffness in patients with chronic kidney disease [93,94].

We found that orally administered EPA significantly suppresses aortic calcification induced in kl/kl mice [18,89]. The reduction of aortic calcification in living mice was detected by a multi-detector CT scan in a clinical situation. The levels of NADPH oxidase-4 (NOX4) mRNA expression and NOX activity in aortic smooth muscle cells isolated from kl/kl mice are higher than those in cells isolated from wild type mice. EPA-supplemented media suppresses NOX4 mRNA upregulation and NOX activity observed in smooth muscle cells from kl/kl mice. Elevated oxidative stress promotes arterial calcification, and NOX is a major source of reactive oxygen species in atherosclerosis [95,96]. EPA reduces NOX activity via free fatty acid receptor 4 (FFAR4), also known as G-protein-coupled receptor 120 (GPR120). Additionally, Klotho downregulates PIT1, a sodium-dependent phosphate transporter, to suppress phosphate uptake, in addition to enhancing phosphaturia. Therefore, a decrease of Klotho leads to mineralization in response to phosphate uptake [16,97]. Even though EPA does not improve hyperphosphatemia, progression of calcification is prevented by EPA in kl/kl mice [18]. This suggests that EPA might suppress phosphate uptake in smooth muscle cells.

### 6.3. Wnt Signaling-Induced Osteogenic Changes in Vascular Smooth Muscle Cells

The Wnt signaling pathway is an evolutionarily conserved pathway regulating cell proliferation and differentiation in development, stem cell renewal, and cancer progression [98,99]. This signaling pathway is also involved in various aging phenotypes including chronic lung diseases, renal fibrosis and arterial calcification [100,101,102,103].

CMV-Msx2 transgenic (CMV-Msx2 Tg^+^) mice fed a high-fat diet exhibit marked cardiovascular calcification involving the aorta and coronary artery [104]. Msx2 suppresses the expression of Dkk1, a Wnt inhibitor, in the aorta. TOPGAL^+^; CMV-Msx2 Tg^+^ mice exhibit augmented aortic LacZ expression, suggesting enhanced Wnt signaling. 

Wnt signaling is over-activated also in kl/kl mice, because Klotho inhibits the binding of Wnt ligands to their receptors [105]. Omega-3 fatty acids suppress Wnt signaling in cancer cells [106,107]. This suggests that EPA may antagonize Wnt signaling also in kl/kl mice and prevent progression of arterial calcification. 

In human aortic smooth muscle cells, activation of Wnt signaling induces the expression of osteogenic genes [78,108]. EPA suppresses not only the expression of these osteogenic genes but also the expression of Wnt signaling marker genes via upregulation of peroxisome proliferator-activating receptor gamma (PPARγ) [78]. PPARγ activates secreted frizzled-related protein 2 (SFRP2) and counteracts vascular calcification induced by Wnt5a in mice [109]. Additionally, EPA upregulates Klotho in the kidney [78]. 

On the other hand, decreased Wnt signaling has emerged as a risk factor for cholesterol accumulation, foam cell formation and atherosclerosis [110]. Therefore, the mechanism is not simple.

### 6.4. Palmitic Acid-Induced Osteogenic Changes in Vascular Smooth Muscle Cells

Stearoyl-CoA desaturase (SCD) enzymes catalyze the conversion of saturated fatty acids to monounsaturated fatty acid, that is, they regulate the intracellular balance of saturated and unsaturated fatty acids [111,112,113]. Mice with smooth muscle cell-specific deletion of Scd1 and Scd2 show severe vascular calcification with increased endoplasmic reticulum stress [114]. Higher dietary intakes of major saturated fatty acids are associated with an increased risk of coronary heart disease [115,116]. Palmitic acid, a saturated long chain fatty acid with a 16-carbon backbone, induces apoptosis, oxidative stress and endoplasmic reticulum stress and increases medial calcification [117]. 

In human aortic smooth muscle cells, EPA suppresses osteoblastic differentiation and mineralization induced by palmitic acid through Acyl-CoA synthetase long chain family member 3 (ACSL3) and nuclear factor-κB (NF-κB) [79]. ACSL3 is highly expressed in vascular smooth muscle cells and macrophages in human non-calcifying and calcifying atherosclerotic plaques from carotid arteries. ACSL3 is an enzyme that converts palmitic acids to palmitoyl-CoA, and the metabolite activates the NF-κB signaling pathway. Palmitic acid upregulates ACSL3 expression prior to osteoblastic gene induction, and EPA suppresses the upregulation of ACSL3 induced by palmitic acid.

### 6.5. Statin-Induced Interleukin 1β (IL-1β)-Mediated Arterial Calcification

IL-1β is an inflammatory cytokine produced by activated macrophages. Statins reduce cardiovascular events, but they promote IL-1β production and consequent plaque calcification and insulin resistance [60,118]. Canakinumab, a therapeutic monoclonal antibody targeting interleukin-1β, inhibits aortic calcification in LDLR^−/−^ mice [119]. In the CANTOS trial, canakinumab significantly reduced high-sensitivity C-reactive protein levels and cardiovascular event rates, and 93.4% of the patients were treated with statins [120]. These results suggest that the addition of drugs suppressing IL-1β to statins may be effective for suppression of statin-induced calcification. EPA suppresses IL-1β expression in rheumatoid arthritis model rats and Thy-1 nephritis model rats [121,122]. 

## 7. Effects of EPA on Arterial Calcification in Clinical Studies

Blood levels of n-3 polyunsaturated fatty acid and omega-3 index and the percentages of EPA and docosahexaenoic acid (DHA) in total fatty acids present in the erythrocyte membrane have been shown to be inversely associated with coronary artery calcification in some observational studies [25,27,28]. On the other hand, we reported that 1.8 g/day EPA does not attenuate progression of coronary artery calcification evaluated by CT in pitavastatin-treated patients with hypercholesterolemia who were asymptomatic for cardiovascular disease during a period of 12 months [26]. Evaluation using optical coherence tomography and intravascular ultrasound showed that the addition of 1.8 g/day EPA to statin therapy does not change the calcified plaque volume, even though it does stabilize and reduce plaques of human coronary arteries [123,124,125]. In an ongoing trial, the EVAPORATE study, changes in coronary plaque, including low-attenuation, fibro-fatty, fibrous, calcified and non-calcified plaque, over a period of 9 to 18 months are being examined in patients treated with statins plus a higher dose of EPA, 4 g/day [11]. The results of interim analysis at nine months, which were presented in 2019 AHA Scientific Sessions (Philadelphia, PA), indicated slowed progression of calcified plaque: EPA, 1% decrease vs. placebo, 9% increase, *p* = 0.001 [126]. These clinical studies were summarized in Table 2. CT, computed tomography; OCT, optical coherence tomography; PCI, percutaneous coronary intervention; IVUS, intravascular ultrasound.

## 8. Future Challenges and Possible Solutions

Unfortunately, there is no clinically established therapy to suppress arterial calcification. We need to think about why EPA does not suppress arterial calcification in a clinical situation even though EPA significantly suppresses arterial calcification in experimental models. The dose might be too low to suppress calcification in patients, because the doses used in experimental animals were very high (the food intake of mice is generally 3–5 g/day and the body weight is generally 20 g; 5% (*w/w*) EPA roughly corresponds to 7.5–12.5 g EPA/kg in mice.) [18,19]. In the JELIS, mean plasma EPA concentrations and the EPA/AA ratio were 170 μg/mL and 1.21, respectively, in patients treated with 1.8 g/day EPA [23]. In the REDUCE-IT, the mean plasma EPA concentration was 144.0 μg/mL in patients treated with 4 g/day EPA [22]. The mean plasma EPA concentrations and EPA/AA ratio were 393.5 μg/mL and 9.15, respectively, in 5% (*w/w*) EPA-fed kl/kl mice [18]. Thus, the serum level of EPA and the EPA/AA ratio were also very high in the experimental study.

Since it is difficult to use such a high dose in a clinical situation, we need to identify a specific target and develop small molecules or oligonucleotide therapeutics that are effective for arterial calcification.

### 8.1. Activation of GPR120 Signaling

GPR120 is a member of the rhodopsin family of G-protein-coupled receptors for long chain fatty acids [127]. GPR120 mediates anti-inflammatory and insulin-sensitizing effects of omega-3 fatty acids [128]. Since EPA suppresses arterial calcification and aortic aneurysm through GPR120, specific activation of GPR120 signaling might be more effective [18,43]. 

### 8.2. Activation of ChemR23 Signaling

Serum resolvin E1 levels are higher in EPA-fed mice [18]. Resolvin E1, a metabolite derived from EPA, exhibits resolving inflammation effects at a low dose and also inhibits phosphate-induced calcification of vascular smooth muscle cells through chemokine like receptor 1 (CMKLR1), also known as ChemR23 [129,130]. Interestingly, chemerin, one of the adipokines, also inhibits phosphate-induced calcification in vascular smooth muscle cells through ChemR23 [131]. Thus, activation of ChemR23 signaling also might be a good target of arterial calcification.

### 8.3. Identification of Other Specific Targets of EPA

Since the inhibitory effect of EPA on arterial calcification has been established in experimental animal models, omics analysis might be useful for identifying specific target molecules or signaling pathways of EPA.

However, the clinical significance of the progression of coronary calcification remains controversial. Progression of coronary calcification is an independent factor of mortality [46]. On the other hand, aggressive lipid lowering with high-dose statins promotes coronary calcification [59]. 

## 9. Summary

In this review, we summarized the effects of EPA on arterial calcification. The proposed mechanisms are summarized in Figure 1. Purified EPA, which has anti-atherogenic effects in patients with cardiovascular disease, prevented arterial calcification in experimental studies. However, the efficacy of EPA for prevention of arterial calcification in patients has not been established and further studies are needed.

## Figures and Tables

**Figure 1 ijms-21-05455-f001:**
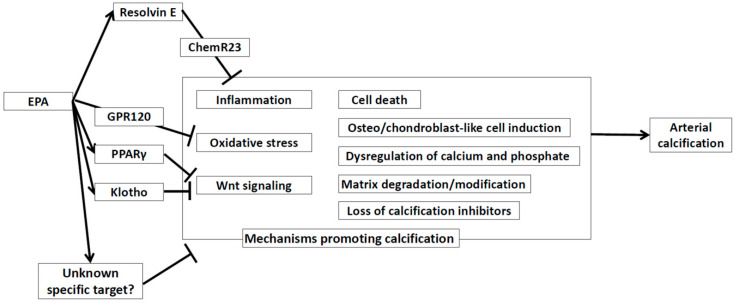
Proposed mechanisms on how EPA affects arterial calcification. EPA: Eicosapentaenoic Acid. ChemR23: Chemerin receptor 23. EPA: eicosapentaenoic acid. GPR120: G-protein coupled receptor 120. PPARγ: Peroxisome Proliferator-Activated Receptor γ.

**Table 1 ijms-21-05455-t001:** Effects of Eicosapentaenoic Acid (EPA) on Arterial Calcification in Experimental Studies.

Induction of Arterial Calcification	Species	In Vivo or In Vitro	Suppression of Calcification	Mechanisms	Reference
Warfarin	Rat	in vivo	Aorta calcification	Suppression of macrophage infiltration, MCP-1 expression and MMP activity in the aorta.	[19]
Klotho deficiency	Mouse	in vivo	Aorta calcification	Suppression of NOX activity inducing oxidative stress via activation of GPR120 signaling in aortic smooth muscle cells.	[18]
Activation of Wnt signaling	Human	in vitro	Osteogenic change	Suppression of Wnt signaling via PPARγ in smooth muscle cells.	[78]
Palmitic acid	Human	in vitro	Osteogenic change Mineralization	Suppression of NF-κB signaling via ACSL3 downregulation in aortic smooth muscle cells.	[79]

**Table 2 ijms-21-05455-t002:** Effects of EPA on Arterial Calcification in Clinical Studies.

Author	Year	Region	Study Patients	Group	EPA Dose	Evaluation Method	Duration	Effect on Calcification
Miyoshi et al. [26]	2018	Japan	Patients with Agatston score 1–999, LDL-C levels ≥140 mg/dL, and no history of atherosclerotic cardiovascular disease	Pitavastatin vs. Pitavastatin plus EPA	1.8 g/day	CT	12 months	No significant difference in annual percent changes in Agatston score and calcium volume score.
Niki et al. [124]	2016	Japan	Statin-treated patients with stable angina scheduled to be treated with PCI	Statin vs. Stain plus EPA	1.8 g/day	IVUS	6 months	No significant difference in percent change in calcium volume.
Watanabe et al. [125]	2017	Japan	Patients with hypercholesterolemia, stable angina or acute coronary syndrome who have received successful PCI with IVUS guidance	Pitavastatin vs. Pitavastatin plus EPA	1.8 g/day	IVUS	6–8 months	No significant difference in calcium volume in non-stenting lesions.
Budoff et al. [11]	ongoing	USA	Statin-treated patients with coronary atherosclerosis, fasting triglyceride levels of 135 to 499 mg/dL, and LDL-C levels of 40 to 115 mg/dL.	Statin vs. Stain plus EPA	4 g/day	CT	18 months	ongoing

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
