# Peer review of "Effects of Eicosapentaenoic Acid on Arterial Calcification"

_ijms, 2020, doi:10.3390/ijms21155455_

Round 1

Reviewer 1 Report

The paper, as far as it goes, is fine. As a recommendation, I would word the Abstract to show your findings, not describe what you are researching. Unless you do this, it is unlikely that your work will be cited anywhere, as researchers will have to read the full paper to discover what you found.

My criticism of the paper is that it seems as though you are writing in a vaccuum as you have provided little context. The Introduction needs to be expanded to discuss this context, as follows:

  1. You need to be very careful when you say coronary artery calcification predicts rupture. On an individual plaque basis, it is in fact protective against rupture; the cuprit plaque in acute coronary syndrome is rarely calcified. There is quite a lot of literature about this paradox and you need to discuss it and show that you know and understand it. The discussion of your later findings needs to reflect this.
  2. You need to be clear on the purpose of giving EPA. Are you trying to prevent CVD or are you trying to lower calcification. 
  3. You need to mention why you are investigating EPA alone, and not with DHA.
  4. You need to discuss other ways in which arterial calcification can be lowered. Vitamin K is one and there is a lot of literature about this as well. Again, bring this into the later discussion.
  5. In Section 4.1: You mention microcalcification but don't discuss macrocalcification. Again, there is a lot of recent literature on this.
  6. In Section 4.2 line 92: This sentence makes no sense. If you are you saying that others have discussed the mechanism then you should expand on this. This is where you bring in the earlier point about calcification being protective against rupture. In this context, the fact that statins increase calcification makes sense.
  7. Section 8: You discuss doses but it would be helpful to translate the experimental dosage into human dosage before you conclude that it is difficult to use such a high dose in a clinical situation.

Author Response

Reviewer 1

We greatly appreciate the reviewer’s comments.

Comment: The paper, as far as it goes, is fine. As a recommendation, I would word the Abstract to show your findings, not describe what you are researching. Unless you do this, it is unlikely that your work will be cited anywhere, as researchers will have to read the full paper to discover what you found.

Response:Thank you for this suggestion. We have added the following description in the Abstract, line 15-17. “We found that EPA significantly suppressed arterial calcification in vitro and in vivo via suppression of inflammatory responses, oxidative stress and Wnt signaling.”

Comment: My criticism of the paper is that it seems as though you are writing in a vacuum as you have provided little context. The Introduction needs to be expanded to discuss this context, as follows:

  1. You need to be very careful when you say coronary artery calcification predicts rupture. On an individual plaque basis, it is in fact protective against rupture; the culprit plaque in acute coronary syndrome is rarely calcified. There is quite a lot of literature about this paradox and you need to discuss it and show that you know and understand it. The discussion of your later findings needs to reflect this.

Response: As the reviewer pointed out, macrocalcifications detected by CT is protective against plaque rupture.

We have deleted “plaque rupture” from the sentence in Introduction, line 24.  We have added the following description in Introduction, paragraph 1, “Coronary artery calcification detected by computed tomography (CT) reflects atherosclerotic plaque burden and predicts cardiovascular events. Additionally, aortic calcification predicts mortality and non-fatal cardiovascular events in dialysis patients. However, macrocalcifications occurring in the intimal plaque during late-stage calcification are believed to stabilize the plaque. On the other hand, microcalcifications occurring during the early-stage calcification correlate with plaque vulnerability that results in myocardial and cerebral infarction, but it is difficult to detect microcalcifications using a clinical CT scan. Medial calcification causes arterial stiffness that is a predictor of coronary heart disease and stroke.”.

Comment: 2. You need to be clear on the purpose of giving EPA. Are you trying to prevent CVD or are you trying to lower calcification.

Response: The purpose is to try to lower calcification and we have added the following sentence in Introduction, paragraph 3, “Thus, we expect that lowering arterial calcification by EPA leads to reduction of cardiovascular events.”.

Comment: 3. You need to mention why you are investigating EPA alone, and not with DHA.

Response: We have added the following description in Introduction, paragraph 3.

“Although docosahexaenoic acid (DHA) also inhibits calcification of vascular cells in vitro, it raises serum LDL-C level that could be a risk of cardiovascular diseases.”

Comment: 4. You need to discuss other ways in which arterial calcification can be lowered. Vitamin K is one and there is a lot of literature about this as well. Again, bring this into the later discussion.

Response: We have added the following description in Introduction, paragraph 2.

“Although there is no clinically established therapy for arterial calcification, some molecules are expected to reduce arterial calcification, e.g. eicosapentaenoic acid (EPA), vitamin K1, magnesium, spironolactone and evolocumab.”.

The roles of vitamin K and evolocumab are described in section 6.1 and 4.2, respectively.

Comment: 5. In Section 4.1: You mention microcalcification but don't discuss macrocalcification. Again, there is a lot of recent literature on this.

Response: We have deleted the following sentence in section 4.1, paragraph 3.

“On the other hand, macrocalcification can be identified using CT and is widely believed to stabilize the plaque.”.

Comment: 6. In Section 4.2 line 92: This sentence makes no sense. If you are you saying that others have discussed the mechanism then you should expand on this. This is where you bring in the earlier point about calcification being protective against rupture. In this context, the fact that statins increase calcification makes sense.

Response: We have deleted the following sentences, “Currently, the mechanism of the increase of arterial calcification by statins have been proposed.” and “However, statin use does not weaken the prognostic utility of coronary artery calcification.”.

Comment: 7. Section 8: You discuss doses but it would be helpful to translate the experimental dosage into human dosage before you conclude that it is difficult to use such a high dose in a clinical situation.

Response: If the weight of a patient is 60 kg, 1.8 or 4 g EPA/day is 0.03 or 0.067 g/kg in human. The food intake of mice is generally 3-5g/day and the body weight is generally 20 g. So, 5% (w/w) EPA corresponds to 7.5-12.5 g EPA/kg in mice. We have added the following description in section 8.

“The food intake of mice is generally 3-5g/day and the body weight is generally 20 g. 5% (w/w) EPA roughly corresponds to 7.5-12.5 g EPA/kg in mice.

Thank you for your consideration of our paper.

Reviewer 2 Report

In this paper, Saito et al. gave an extensive review on the effects of eicosapentaenoic acid on arterial calcification. It is well done. Few minor comments: 

-Abstract: instead of stating "there is no evidence showing the effect of EPA on arterial calcification in clinical situation," may be the authors can state "there is so far lack of evidence showing..." 

-Page 2, line 55, need to clarify what "control" was. I think it was statin alone instead of placebo

-In table 1, please include references for the studies

-Page 4, 6.1, please consider moving the sentence "warfarin antagonize vitamin K and blocks gamma-carboxylation of MGP and induces vascular calcification" to the beginning of the paragraph. 

-Table 2, please add references

-lastly, a figure or a schematic diagram summarizing the proposed mechanisms on how EPA may affect arterial calcification would be helpful for the readers

Author Response

Reviewer 2

We greatly appreciate the reviewer’s comments.

Comment: In this paper, Saito et al. gave an extensive review on the effects of eicosapentaenoic acid on arterial calcification. It is well done. Few minor comments:

-Abstract: instead of stating "there is no evidence showing the effect of EPA on arterial calcification in clinical situation," may be the authors can state "there is so far lack of evidence showing..."

Response: Thank you for this comment. We have corrected that as the reviewer suggested.

Comment: -Page 2, line 55, need to clarify what "control" was. I think it was statin alone instead of placebo

Response:As the reviewer said, the control group was statin alone group. We have corrected that as following description.

JELIS (statin plus EPA, 2.8% vs. statin alone, 3.5%)

Comment: -In table 1, please include references for the studies

Response:We have added references in Table 1.

Comment:-Page 4, 6.1, please consider moving the sentence "warfarin antagonize vitamin K and blocks gammacarboxylation of MGP and induces vascular calcification" to the beginning of the paragraph.

Response:Thank you for this suggestion. We have moved this in the beginning of the paragraph.

Comment:-Table 2, please add references

Response:We have added references in Table 2.

Comment:-lastly, a figure or a schematic diagram summarizing the proposed mechanisms on how EPA may affect arterial calcification would be helpful for the readers

Response: Thank you for this suggestion. We have added a figure summarizing the proposed mechanisms (Figure 1).

Thank you for your consideration of our paper.